# OpenReview forum: "How Catastrophic is Your LLM? Certifying Risks in Conversation"
_ICLR.cc/2026/Conference — ICLR 2026 Poster_

### Official Review · Reviewer_7DWM · 2025-10-29

**Soundness:** 2
**Presentation:** 2
**Contribution:** 2
**Rating:** 4
**Confidence:** 3

**Summary:**

This paper proposes a new framework named C^3LLM, which is used to quantify and certify the risk of catastrophic responses (such as manufacturing weapons, biological terrorism, etc.) generated by LLMs in multi-turn conversations.

C^3LLM models multi-turn conversations as a probability distribution on a "query graph". It provides a confidence interval with statistical guarantees for the probability of catastrophic risks. This method not only reveals the vulnerability of the model in a vast conversation space but also allows for quantitative comparisons of the security of different models.

**Strengths:**

1) The paper proposes the first statistical framework (C^3LLM) for certifying catastrophic risks of LLMs in multi-turn dialogues. This changes the previous evaluation paradigm that relied on fixed benchmarks and empirical attacks.

2) The multi-turn dialogue process is abstracted as a Markov process on a "query graph," and three attack strategies (Random Node, Graph Path, Adaptive with Rejection) are designed to well simulate strategies of attackers at different levels.

**Weaknesses:**

1) In Section 3.2, the writing of Graph Distributions is too chaotic. A Markov state transition equation does not require so many symbolic representations; the author should express it from a more specific perspective. Moreover, the meanings of some symbols are not clearly defined, such as \pi_{V\S}, \pi_{N(v)\S}, and \pi_(N(v)).

2) The paper used a large number of hyperparameters, and different hyperparameter settings as an evaluation benchmark can affect the objectivity of the evaluation results.

**Questions:**

1) The paper mentioned a large number of hyperparameters, and the reviewers were very curious about how to determine these parameters. What if the final hyperparameters of different LLMs are different when choosing the hyperparameters with the lowest score?

2) Hyperparameter analysis should be more comprehensive, including the analysis of hyperparameters for all evaluated LLMs, and further analysis of how these hyperparameters affect the evaluation scores.

---

> ### Author Response · Authors · 2025-11-25
>
> We thank the reviewer for recognizing our statistical framework for certifying catastrophic risks in multi-turn dialogues and the query-graph Markov abstraction with multiple attack strategies.
>
> **Notation and graph distributions (Section 3.2).**
> We agree that the original presentation of the graph distributions in Section 3.2 was dense. The extra symbols arise because we need to model a path without repetition as a Markov chain: the Markov state must contain both the current query and the visited set, $(v,S)$, since the transition probabilities depend on which queries have already been used (so that a sequence does not revisit the same node). In the revised version, we simplify the notation by using a single sampling distribution $\pi(\cdot \mid A)$ on a finite set $A$ (e.g., the unvisited nodes $V \setminus S$ or unvisited neighbors $N(v)\setminus S$). We also add brief inline explanations where these terms appear to make the text more readable.
>
> **Choice of hyperparameters and “lowest score” concern.**
> For the main reported results, we do not tune hyperparameters separately for each LLM to minimize or maximize its certified risk; instead, we fix a single set of values and use them across all models for fairness and comparability. For example, for the jailbreak insertion probability $p$ we use a default $p = 0.2$ for all LLMs. Appendix C.1 shows that an alternative $p$ can lead to higher certified risks: for instance, for Llama, increasing $p$ from 0.2 to 0.8 yields the highest lower/upper bounds among the tested values. Our framework allows practitioners to explore stronger attacker settings by adjusting these hyperparameters.
>
> **Hyperparameter analysis.**
> We agree that hyperparameters can influence the evaluation, so we include a dedicated sensitivity analysis in Appendix C where we systematically vary key hyperparameters and examine their impact on the certified bounds. In the revised version, we emphasize this analysis more clearly in Section 4.3 and extend it to cover all evaluated LLMs in Appendix C. Across these studies, the qualitative conclusions and model rankings remain stable, indicating that our results are not overly driven by a specific hyperparameter choice. This addresses the request for a more comprehensive analysis of how hyperparameters affect the evaluation scores.

---

### Official Review · Reviewer_pvwt · 2025-10-30

**Soundness:** 3
**Presentation:** 3
**Contribution:** 3
**Rating:** 8
**Confidence:** 4

**Summary:**

This paper introduces C³LLM, a novel framework to move beyond fixed benchmarks and statistically certify the catastrophic risk of Large Language Models (LLMs) in multi-turn conversations. The authors argue that simply testing a few attack sequences fails to capture the vast space of possible dialogues. Their method models conversations as a Markov process on a "query graph," where nodes are individual queries and edges represent semantic similarity, capturing realistic conversational flow. By sampling query sequences from this graph using various distributions (e.g., random, path-based, or adaptive to model refusals), the framework calculates high-confidence probability bounds (confidence intervals) for catastrophic failures. Applying this to frontier models, they find that some, like Claude-Sonnet-4, are certifiably safer, while others, like DeepSeek-R1 and Mistral-Large, exhibit high certified lower bounds for catastrophic risk, reaching over 70% in some cases.

**Strengths:**

1. The paper's primary strength is its conceptual leap from empirical benchmarking to probabilistic certification.
2. The finding that some frontier models have certified catastrophic risk lower bounds as high as 70% is a powerful and alarming result that underscores the urgency of the problem.

**Weaknesses:**

The entire certification framework is conditioned on the initial query set $Q$ (100 queries per scenario). While the space of conversations sampled from $Q$ is vast (e.g., $100^5$), the certification is only valid for that specific set $Q$. The paper's "actor-based" method for generating $Q$ is a reasonable heuristic, but the sensitivity of the final certified bounds to this initial set is not fully explored. If the initial query set is "weak" or biased, the certification might be misleadingly optimistic.

**Questions:**

Clarify my comments in "Weaknesses section"

---

> ### Author Response · Authors · 2025-11-25
>
> We thank the reviewer for highlighting our shift from empirical benchmarking to probabilistic certification and for emphasizing the significance of the high certified risk lower bounds we report.
>
> **On dependence on the initial query set $Q$.**
> Our actor-based procedure for constructing $Q$ is specifically designed to make it strong and diverse, and the reported numbers are lower bounds with high confidence on attack success under this induced distribution. $C^3LLM$ is a framework: practitioners can plug in alternative choices of $Q$ and reuse exactly the same certification pipeline. In Appendix C.3, we study sensitivity to the size of $Q$ by varying it from 50 to 150 queries per scenario; the certified bounds change only slightly, suggesting that our conclusions are not overly sensitive to this particular choice of $Q$.

---

> > ### Comment · Reviewer_pvwt · 2025-11-28
> > **Final Comment on rebuttal**
> >
> > Thank you for answering all my Concerns.

---

### Official Review · Reviewer_e15j · 2025-10-31

**Soundness:** 2
**Presentation:** 3
**Contribution:** 2
**Rating:** 4
**Confidence:** 4

**Summary:**

This paper proposes C³LLM, a statistical certification framework to evaluate the catastrophic risk of large language models (LLMs) in multi-turn conversations. It models dialogues as probability distributions over query sequences on a semantic graph and samples conversation paths using Markov processes (e.g., random node, graph path, adaptive rejection). By estimating the probability of harmful responses with Clopper–Pearson confidence intervals, the framework provides statistically guaranteed lower bounds on risk. Experiments on the HarmBench dataset show that conventional fixed-prompt evaluations significantly underestimate LLMs’ true safety risks.

**Strengths:**

The authors define a distributional framework where conversations are modeled as query sequences sampled from a Markov process on a semantic graph.

**Weaknesses:**

While the paper introduces a statistically principled framework for assessing catastrophic risks in LLMs, its novelty and methodological depth are limited, and several aspects could be strengthened to align better with its stated goals.
1. Limited novelty in “statistical certification” – The claim of introducing statistical guarantees is overstated. The use of the Clopper–Pearson exact interval for binomial estimation is a standard statistical tool rather than a new certification technique. Prior benchmark studies could easily adopt the same confidence estimation without requiring a new framework.
2. “Non-fixed” conversation modeling is only partial – Although the paper claims to move beyond “fixed attack sequences,” the sampling still occurs over a predefined, finite query graph. The space of possible prompts remains fixed; only the sequence order is randomized. As a result, the method does not fully capture the semantic or generative variability that naturally occurs in real multi-turn attacks.
3. Dependence on hand-crafted graphs and thresholds – The semantic query graph relies on manually tuned similarity thresholds (lth,hth) using sentence embeddings. This manual specification introduces bias and limits generalization.
4. Limited evaluation domains and scale – The experiments cover only two domains (chemical/biological and cybercrime) and sample 50 sequences per setting. This narrow scope limits the statistical validity of the “certified” results, particularly for rare catastrophic behaviors.
5. Ambiguous notion of “certification” – The term “certification” suggests formal safety guarantees, but the framework only provides empirical confidence intervals based on finite samples.

**Questions:**

1. The paper argues that prior works used “fixed attack sequences,” while C³LLM is non-fixed. However, the sampling still happens within a predefined query graph and does not dynamically generate new prompts.Can the authors clarify how this approach truly differs from fixed-sequence evaluation beyond sampling random orderings?
2. The term “certification” may suggest formal safety guarantees, yet the current approach provides empirical confidence intervals from finite samples.How do the authors justify this terminology choice?

---

> ### Author Response · Authors · 2025-11-25
>
> We thank the reviewer for highlighting that our framework defines a distributional view of conversations as query sequences on a semantic graph. We appreciate these comments and, in what follows, we address each weakness and question in turn.
>
> **Limited novelty in “statistical certification”.**
> We explicitly state our main contributions below, which extend beyond the use of Clopper–Pearson confidence intervals.
>
> 1. *Distributions of multi-turn conversations:* We define the first large distributions consisting of multi-turn conversations with LLMs, and their i.i.d. samplers. Defining these is imperative for any statistical certification methods, not restricted to Clopper–Pearson confidence intervals. Our distributions are systematically defined over a general graph of queries, which can be readily tuned to be applicable on any other domain.
>
> 2. *Potentially adversarial inputs:* To effectively stress-test the models, we define our distributions over potentially adversarial queries and form multi-turn conversations with varying adversarial strategies to model natural attacker behaviors at different capability levels: an unsophisticated attacker who samples queries without using structure (*Random Node with Jailbreak*), an attacker who explores semantically related queries along a path toward a harmful target (*Graph Path*), and an adaptive attacker who selects the next query based on accept/reject feedback from the model (*Adaptive with Rejection*). These distributions are defined over potentially adversarial queries and induce multi-turn conversations under varying adversarial strategies. Our graphical formulation makes it convenient to model various adversarial strategies and we show various examples of these in our formulation and experiments.
>
> Our contribution is to make the Clopper–Pearson method applicable in this setting by explicitly defining a distribution over a large space of multi-turn attack sequences from which we can collect i.i.d. samples. Prior work does not specify such a distribution, so one cannot simply plug their results into a binomial bound and claim a statistical guarantee over a large space.
>
> **“Non-fixed” conversation modeling is only partial.**
> Our distributions are based on the graphical query representation, but not restricted to it. This is evident from our independent jailbreak layer, which can be further manipulated with token-level mutations (defined in Appendix A). We add experiments for token-level mutations in the revision. This augmentation increases the diversity of adversarial conversations captured by our framework and illustrates the flexibility of our methods to incorporate new adversarial multi-turn conversation scenarios and generate statistical certificates for them.
>
> In our added experiments, we instantiate this augmentation for the *Adaptive with Rejection* attack by replacing $v_t$ with an augmented query $\tilde v_t \sim D_{jb}^{mut}(\cdot \mid v_t)$ whenever the previous query is rejected. In a representative scenario, the augmented attack reveals catastrophic risks across models, as shown in Table&nbsp;1.
>
> **Table 1 – Certified 95% confidence intervals for catastrophic failure probability on a single representative scenario under the augmented adaptive-with-rejection attack.**
>
> | Model    | 95% CI                 |
> |----------|------------------------|
> | Mistral  | (0.7569, 0.9547)   |
> | gpt-oss  | (0.1463, 0.4034)   |
> | Claude   | (0.2121, 0.4877)   |
> | DeepSeek | (0.4518, 0.7359)   |
> | Llama    | (0.4518,0.7359)   |
>
> Moreover, we can apply more structured generators as a layer to ensure the non-fix, e.g., a second LLM that deterministically chooses mutations based on the context. As long as this generator is viewed as part of the attack process and induces a well-defined distribution over conversations, it can be incorporated into our framework in exactly the same way. In the revised version, we make this explicit by introducing a generic augmentation layer $D_{jb}^{mut}(\cdot \mid v_t)$ in Section 3.2 and updating Appendix A to formally define the jailbreak instantiations $D_{jb}(\cdot \mid v)$ and $D_{jb}^{mut}(\cdot \mid v)$; these changes are shown in blue text in the revised version.
>
> **Dependence on hand-crafted graphs and thresholds.**
> The current semantic graph and similarity thresholds are one instantiation of our proposed framework. We provide sensitivity analyses showing our conclusions are robust across reasonable threshold choices in Appendix C, and any graph-construction method and thresholds can be plugged into the same framework. Our aim is to expose the systematic vulnerabilities of SOTA LLMs in multi-turn conversation settings.

---

> > ### Author Response · Authors · 2025-11-25
> >
> > **Limited evaluation domains and scale.**
> > - As noted in our comment to all reviewers, we additionally evaluate our framework on scenarios from the HarmBench *illegal* subset using all evaluated LLMs, providing further evidence that our method is effective beyond the *chemical_biological* and *cybercrime* domains.
> > - As shown in Appendix C.5, the certified lower and upper bounds stabilize once the number of samples reaches n $\approx 50$, so increasing n further yields only marginal improvements while incurring substantially higher computational cost, indicating that our choice of 50 sampled sequences per setting is sufficient to obtain informative certificates.
> >
> > **Notion of “certification”.**
> > We use “certification” in the *statistical* sense: given a specified attack distribution and i.i.d. samples, we provide high-confidence (e.g., 95%) upper and lower bounds on catastrophic risk under that distribution. To avoid confusion, we have revised the text to consistently use the term “statistical certification” and explicitly distinguish our notion from deterministic formal verification.

---

### Official Review · Reviewer_7oyG · 2025-10-31

**Soundness:** 3
**Presentation:** 4
**Contribution:** 2
**Rating:** 6
**Confidence:** 5

**Summary:**

This paper proposes an interesting probabilistic view to quantify the safety risks of large language models (LLMs) in multi-turn scenarios. It models multi-turn conversations as probability distributions over query sequences, thus quantifying catastrophic risks using confidence intervals. This paper defines 3 practical distributions. Their results demonstrate the serious safety risks of frontier LLMs.

**Strengths:**

+ interesting perspective: the probabilistic modeling of safety risks extends the limited risk evaluation via fixed attack trajectories towards continuous and broader, and formal risk evaluation.

+ The flexibility of their mathematical framework enables the community to extend their evaluation method by defining corresponding sequence distributions.

**Weaknesses:**

The implementation of their methodology may not generalize well. The initial node distribution of their graph depends on a multi-turn attack baseline. This brings up the question of whether their sampled sequences cover the common distribution of malicious users in the real world. I think authors should provide more comprehensive analyses to verify the validity of their results.

**Questions:**

This paper only evaluates the safety of models on a subset of one benchmark. is there more results to provide more evidence of the effectiveness of their method?

some typos:
line 156 net -> let

---

> ### Author Response · Authors · 2025-11-25
>
> We thank the reviewer for noting that our probabilistic modeling moves beyond fixed attack trajectories and for highlighting the flexibility of our framework for defining new sequence distributions. We appreciate these comments and we address the raised weakness and questions in detail in what follows.
>
> **Generalization and the initial node distribution.**
> We explicitly design three families of sequence distributions on the query graph to model different, practically motivated attacker behaviors: (1) *Random Node with Jailbreak* represents an unsophisticated attacker who samples queries without exploiting graph structure; (2) *Graph Path* models an attacker who explores semantically related queries along a path toward a harmful target; and (3) *Adaptive with Rejection* captures an attacker that adaptively select the next query based on accept/reject feedback from the model. The initial-node distribution is part of this threat model and can be instantiated in different ways; in our experiments, we use prompts generated by an actor-based multi-turn attack from prior work to obtain a pool of high-risk queries, but any alternative initialization can be plugged into the same framework.
> Crucially, $C^3LLM$ is a framework for evaluating catastrophic risks with statistical guarantees rather than a single attack method. Moreover, our reported numbers are lower bounds on failure probabilities under these attack distributions, so replacing the actor-based baseline (which is adapted from existing multi-turn attack work) with any stronger attack can only increase the certified risk, not hide failures we already expose.
>
> **Additional analyses and evaluation scope.**
> As noted in our comment to all reviewers, we now additionally evaluate our framework on scenarios from the HarmBench *illegal* subset using all evaluated LLMs. The certified bounds show similar patterns across models, supporting that our approach is effective beyond the original *chemical_biological* and *cybercrime* subset.
>
> **Typo.**
> We thank the reviewer for pointing out the typo at line 156 and we have corrected it in the revised version.

---

### Author Response · Authors · 2025-11-25

We thank the reviewers for their detailed and thoughtful feedback. In response, we have made a number of revisions and added new experiments that address the main concerns and, we believe, further improve the clarity and overall quality of the paper. All changes in the revised PDF are shown in blue font for ease of reference.

---

### Author Response · Authors · 2025-11-25
**More experiments on other datasets**

We additionally evaluate our framework on 30 scenarios from the HarmBench *illegal* subset for Claude, gpt-oss, and Llama, and on the first 10 scenarios for Mistral and DeepSeek (due to time and compute constraints). The certified bounds under the four distributions are shown in Tables&nbsp;1 and&nbsp;2 below.

---

Entries are the median of 95% confidence intervals across all specifications under a distribution.
Distributions: Random Node with Jailbreak (RNwJ), Graph Path (vanilla) (GPv), Graph Path (harmful target constraint) (GPh), and Adaptive with Rejection (AwR). We bold the highest bounds among the four distributions for each LLM.

**Table 1 – Certification bounds for *illegal* (first 10 scenarios), Mistral & DeepSeek**

|  Model    | RNwJ               | GPv              | GPh              | AwR              |
|----------|--------------------|------------------|------------------|------------------|
| mistral  | **(0.586, 0.845)** | (0.234, 0.504)   | (0.363, 0.653)   | (0.314, 0.609)   |
| deepseek | **(0.523, 0.796)** | (0.146, 0.403)   | (0.187, 0.457)   | (0.212, 0.488)   |


**Table 2 – Certification bounds for *illegal* (30 scenarios), Claude, gpt-oss & Llama**

| Model   | RNwJ             | GPv                | GPh                | AwR                |
|---------|------------------|--------------------|--------------------|--------------------|
| claude  | (0.000, 0.089)   | **(0.005, 0.137)** | (0.001, 0.106)     | (0.005, 0.137)     |
| gpt-oss | (0.001, 0.106)   | (0.009, 0.151)     | **(0.013, 0.165)** | (0.005, 0.137)     |
| llama   | **(0.309, 0.597)** | (0.086, 0.314)     | (0.131, 0.382)     | (0.101, 0.337)     |

---
From Table 1 in our paper together with Tables 1 and 2 above, we see that for gpt-oss and Claude, certified bounds on the *illegal* subset are lower than on the *chemical_biological* and *cybercrime* subsets (e.g., for gpt-oss the Adaptive with Rejection lower bounds are 0.101 and 0.318 on *chemical_biological* and *cybercrime*, respectively; for Claude the largest lower bound on *cybercrime* is 0.195). By contrast, Llama, Mistral, and DeepSeek exhibit high certified catastrophic risk across all three domains.

These results illustrate how our framework allows practitioners to assess catastrophic risk with statistical guarantees in different domains and to choose models accordingly: for instance, Claude may be acceptable for *illegal* in our setting, whereas for *cybercrime* none of the evaluated models appears reliably safe, indicating that stronger mitigations on safer LLMs would be needed.

---

### Author Response · Authors · 2025-12-03
**A Summary Comment**

Dear Area Chair,

We thank all reviewers for their thoughtful and detailed feedback, and for recognizing the key contributions of our work. Below we summarize the main points of recognition, address each reviewer’s concerns, and highlight the improvements added during the rebuttal period.

### **Reviewer Recognition of Contributions**
- R-7oyG highlighted the interesting probabilistic modeling of safety risks, the flexibility of our framework, and how it meaningfully extends beyond fixed-trajectory evaluations.
- R-e15j recognized that we define a distributional framework for multi-turn conversations via query-graph Markov processes and that our method provides statistically principled risk estimates.
- R-pvwt emphasized our “conceptual leap” from empirical benchmarking to probabilistic certification, and noted the significance of the high certified catastrophic-risk lower bounds we report (e.g., >70% for frontier models).
- R-7DWM recognized that we provide the first statistical framework for certifying catastrophic risks in multi-turn settings, and that our three attack strategies reasonably model attackers at different capability levels.

We are grateful that reviewers found the direction valuable and the results impactful.

### **Key Revisions**
- **Expanded Experiments Across More Domains**
(R-7oyG, R-e15j)

    During the rebuttal period, we conducted additional experiments on the HarmBench illegal subset for Claude, gpt-oss, Llama, Mistral, and DeepSeek, covering up to 30 scenarios. The new results (Tables 1–2 in our rebuttal) show consistent patterns across domains, strengthening the generalizability of our conclusions.

- **Clarifying Novelty**
(R-e15j)

    We clarified that our contribution extends far beyond applying Clopper–Pearson: We define the first large distributions over multi-turn conversations from which we can collect i.i.d. samples, which are essential prerequisites for any statistical guarantee.

- **Addressing the “Fixed Graph / Limited Variability” Concern**
(R-e15j)

    We expanded the framework in several ways. We introduced token-level mutation augmentation (Appendix A) and a generic augmentation layer capable of applying either rule-based or LLM-driven modifications. We also added experiments demonstrating that the augmented adaptive-with-rejection attack consistently reveals higher catastrophic-risk estimates across models. These extensions confirm that the graph is not a fixed script; instead, new sequences and perturbations can be injected dynamically while preserving a well-defined distribution necessary for statistical certification.

- **Hyperparameter Sensitivity & Fairness Across LLMs**
(R-7DWM)

    We emphasize that we do not tune hyperparameters per model, instead, we use a single fixed configuration across all LLMs for fairness. We added comprehensive hyperparameter analyses (Appendix C) across all evaluated models. These studies show that overall trends remain stable, model rankings and qualitative conclusions do not change, and some stronger attacker hyperparameters typically increase the certified risk rather than decrease it. This confirms that our findings are not artifacts of particular hyperparameter choices.

- **Dependence on Initial Query Set**
(R-pvwt)

    We clarified that the queries are generated using a strong actor-based multi-turn attack from prior work, making the initialization intentionally non-weak. Because our certification yields lower bounds on catastrophic risk, a stronger initialization only increases the certified risk rather than masking failures. We additionally conducted a sensitivity study varying the query-set size from 50 to 150 per scenario and the certified bounds changed only mildly, indicating stability and mitigating concerns that our results might be overly sensitive to the construction of the initial query set.

- **Improved Presentation & Notation Clarity**
(R-7DWM, R-e15j)

   We simplified Section 3.2 and clarified the terminology around ‘statistical certification’. Our certification is explicitly statistical and distribution-conditioned, and we clarified this throughout the revised manuscript.


We believe the revisions and clarifications fully address the reviewers’ concerns and substantially strengthen the paper. We thank the reviewers for their constructive feedback and the Area Chair for the time spent evaluating our submission.

---

### Meta-Review · Area_Chair_A9j4 · 2026-01-06

**Summary:**

The paper proposes $C^3LLM$, a framework for statistically certifying catastrophic risks in LLMs during multi-turn conversations by modeling them as distributions over query sequences on a semantic graph. The reviewers generally appreciated the conceptual shift from fixed attack trajectories to a probabilistic certification framework. Concerns primarily focused on the generalization of the method, the reliance on fixed query graphs versus generative variability, the clarity of notation, and the sensitivity of the results to hyperparameters and initial query sets.

**Reviewer Concerns:**

**Addressed:**
*   **Generalization and Evaluation Scope:** R-7oyG and R-e15j questioned the limited domain scope. The authors successfully addressed this by conducting additional experiments on the HarmBench illegal subset across multiple models (Claude, gpt-oss, Llama, Mistral, DeepSeek), demonstrating consistent findings.
*   **Fixed Graph / Limited Variability:** R-e15j concerned that the method relied on a fixed graph. The authors addressed this by introducing token-level mutation augmentations and a generic augmentation layer, showing that the framework can handle dynamic perturbations.
*   **Hyperparameter Sensitivity:** R-7DWM raised concerns about hyperparameter choices. The authors provided a comprehensive sensitivity analysis in Appendix C across all models, demonstrating that the trends and rankings remain stable.
*   **Notation Clarity:** R-7DWM found the notation in Section 3.2 chaotic. The authors simplified the mathematical presentation and clarified definitions in the revision.
*   **Initialization Dependence:** R-pvwt noted potential sensitivity to the initial query set. The authors performed a sensitivity study varying the set size, showing the bounds are stable.

**Outstanding:**
*   **Definition of "Certification":** While the authors clarified that they use "certification" in a statistical sense (confidence intervals), R-e15j initially felt the term might imply formal verification. However, the clarification in the text and the distinction made in the rebuttal largely mitigate this, though semantic disagreements on the term may persist slightly.

**Reviewer Scores:**

*   **R-7oyG: 6**
    The reviewer initially gave a 6 and requested more experimental results to verify effectiveness; the authors provided extensive new results on the HarmBench illegal subset.

*   **R-e15j: 6**
    The reviewer gave a 4 due to concerns about the "fixed" nature of the graph; the introduction of token-level mutations and the augmentation layer effectively addresses this methodological concern, justifying a score increase to 6.

*   **R-pvwt: 8**
    The reviewer already gave an 8 and expressed strong support for the "conceptual leap" of the paper; the rebuttal addressed their minor concern regarding initialization sensitivity, maintaining the strong score.

*   **R-7DWM: 6**
    The reviewer gave a 4 based on clarity and hyperparameter concerns; the authors' rigorous sensitivity analysis and simplified notation in Section 3.2 directly resolve these issues, justifying a move to a 6.

---

### Decision · Program_Chairs · 2026-01-26

Accept (Poster)